# Cohort profile: a national, population-based cohort of children born after assisted conception in the UK (1992–2009): methodology and birthweight analysis

Mitana Purkayastha [1] Stephen A Roberts,[2] Julian Gardiner [1,3] Daniel R Brison,[4] Scott M Nelson,[5] Deborah Lawlor,[6] Barbara Luke,[7] Alastair Sutcliffe[1]

For numbered affiliations see end of article.

**Correspondence to**
Dr Mitana Purkayastha;
m.purkayastha@ucl.ac.uk

## ABSTRACT

**Purpose** To generate a large cohort of children born after assisted reproductive technology (ART) in the UK between 1992 and 2009, their naturally conceived siblings (NCS) and matched naturally conceived population (NCP) controls and linking this with health outcome data to allow exploration of the effects of ART. The effects of fresh and frozen embryo transfer on birth weight (BW) were analysed to test the validity of the cohort.

**Participants** Children recorded on the Human Fertilisation and Embryology Authority (HFEA) register as being born after ART between 1992 and 2009, their NCS and matched NCP controls linked to Office for National Statistics birth registration dataset (HFEA-ONS cohort). This cohort was further linked to the UK Hospital Episode Statistics database to allow monitoring of the child's post-natal health outcomes up to 2015 (HFEA-ONS-HES subcohort).

**Findings to date** The HFEA-ONS cohort consisted of 75 348 children born after non-donor ART carried out in the UK between 1 April 1992 and 31 July 2009 and successfully linked to birth registration records, 14 763 NCS and 164 823 matched NCP controls. The HFEA-ONS-HES subcohort included 63 877 ART, 11 343 NCS and 127 544 matched NCP controls further linked to health outcome data. The exemplar analysis showed that children born after fresh embryo transfers were lighter (BW difference: −131 g, 95% CI: −140 to −123) and those born after frozen embryo transfers were heavier (BW difference: 35 g, 95% CI: 19 to 52) than the NCP controls. The within-sibling analyses were directionally consistent with the population control analyses, but attenuated markedly for the fresh versus natural conception (BW difference: −54 g; 95% CI: −72 to −36) and increased markedly for the frozen versus natural conception (BW difference: 152 g; 95% CI: 113 to 190) analyses.

**Future plans** To use this cohort to explore the relationship between ART conception and short-term and long-term health outcomes in offspring.

## INTRODUCTION

Assisted reproductive technology (ART) usage has increased annually since the first live birth in 1978, with over 8 million children born after ART globally.[1] This increased utilisation of ART has occurred concurrently with developments in our understanding of the impact of the early life environment on long-term health[2–4] and most families with ART conceived children reported potential general health risks to their children as their paramount concern,[5 6] resulting in a concomitant increase in focus on the potential adverse short-term and long-term effects on offspring.

The inability to distinguish the relative importance of the effects of ART treatment factors and parental subfertility is a common

### Strengths and limitations of this study

► Meticulous linkage of robust, routinely collected administrative health data to yield a large cohort that is nationally unique, thus increasing the generalisability, accuracy and precision of results from subsequent analyses.

► Linkage to the hospital admissions and outpatient database provides long-term mortality and morbidity outcome data on offspring for use in longitudinal research, policy planning and strategic development.

► Identification of naturally conceived siblings as well as matched naturally conceived population controls allows exploration of the association of assisted reproductive technology (ART) with adverse offspring outcomes while accounting for parental factors related to subfertility, which may confound these associations.

► Comparison of findings between the two approaches (ART vs naturally conceived population controls and ART vs naturally conceived siblings) mentioned above increases confidence in findings.

► The validity of the cohort was tested by means of an exemplar analysis.

limitation of many ART follow-up studies.[7–9] This issue has been addressed in prospective cohorts by identifying control populations of children born naturally to parents with established subfertility (different from infertility in terms of the time of unwanted non-conception).[10] One such well-known cohort provided the first evidence of differences in blood pressure and growth parameters in ART children.[11 12] Further evidence comes from within-ART comparisons showing that early life environment (eg, embryo culture medium composition) is associated with changes in fetal growth, birth weight (BW) and child growth,[13 14] and studies of embryo cryopreservation have made a similar point more recently.[15] These small studies have been essential in both directing and validating the associations identified from large population studies, including prospective cohort studies of ART conceptions only (eg, a Swedish study that followed-up 30 959 children born after ART)[16] or large, record linkage studies (eg, one South Australian record linkage study compared 6163 ART and 302 811 naturally conceived (NC) offspring, while another study in Denmark compared 33 139 ART children to 555 828 NC children from the population).[17 18] The Committee of Nordic ART and Safety (CoNARTaS), created in 2008, utilised medical registry data from Denmark, Finland, Norway and Sweden to establish a large cohort of children conceived after ART or natural conception as well as women with at least one delivery after ART or natural conception, with the aim of examining long-term health outcomes in children born after ART.[19–21]

While these large studies using conventional multivariable approaches to explore the association of ART with adverse offspring outcomes are essential in order to extrapolate effect sizes and risk estimates to the general ART population,[22–24] they cannot fully account for parental factors related to subfertility, which may confound these associations. This has been addressed in record linkage studies that used within-sibling analyses (where comparisons are made between ART and their NC siblings) to better control for factors related to subfertility and other family confounders under the assumption that these parental factors would be the same (or very similar) within sibling groups.[23 25] However, such analyses typically have lower statistical power due to restricted numbers and can be biased if there is individual level confounding, and the subset of ART children with siblings may not be representative of the ART population as a whole. Therefore, a comparison of the two approaches is valuable as similar results from both would increase confidence in findings.[25–27]

In the UK, birth rates from in vitro fertilisation treatment have increased by over 85% since 1991, with around one in three treatment cycles now resulting in a birth for patients under 35.[28] It has been compulsory for every fertility clinic in the UK to report details of all treatments carried out to the Human Fertilisation and Embryology Authority (HFEA) since its inception in August 1991.[29–31] Due to the mandatory nature of reporting, the completeness and quality of data related to some couple characteristics (eg, age, duration of infertility) and ART procedures are high, enabling research analysing treatment outcomes including success rates and perinatal outcomes such as gestation, BW and congenital anomalies.[32–37] However, the quality of perinatal outcome data available on the HFEA register is questionable as it is patient reported via the ART clinics.[38] Hann et al[39] showed that 1 in 15 (6%) BWs recorded on the HFEA register was incorrect compared with data recorded directly by the delivering maternity unit. Moreover, the HFEA register contains no information on the health outcomes of children beyond the immediate perinatal period.

A change in the law in 2009[40] made it possible for researchers to use patient identifying data contained in the HFEA register from 1991 to 2009 for the purpose of linkage to offspring health outcome databases, thus allowing the largest ever population studies of ART childhood cancer[41] and early child growth.[39] However, further studies have been limited by extensive data governance requirements and high associated costs, and prospective monitoring of outcomes in this population as they progress from childhood into adulthood has been prevented thus far by the paucity of identifiable information necessary for a variety of linkages.

Consequently, the primary objective of this study was to substantially enhance the research value of the HFEA register by utilising electronic record linkage methodology to establish a cohort of children consisting of those born after ART in the UK between 1992 and 2009, their naturally conceived siblings (NCS) and matched naturally conceived population (NCP) controls to allow exploration of outcomes present on the birth registration dataset and subsequent linkage to other datasets. The secondary objective was to create a subcohort of children born after ART in the UK, NCS and matched NCP controls and linking this with information on their postnatal health outcomes up to 2015. These databases will be made available to all researchers (subject to necessary approvals), allowing more precise risk estimates of associations of ART with many potential childhood (and maternal) outcomes. Finally, the third objective of this study was to validate the cohort and subcohort produced by comparing the effects of fresh and frozen embryo transfer versus natural conception on singleton BW. The hypothesis being tested was that children born after frozen embryo transfers would be heavier and those born after fresh embryo transfers would be lighter than those that were NC. Our previous linkage from the HFEA 1991–2009 cohort to maternity and child growth records confirmed evidence in the field that children born after frozen embryo transfers are heavier and those born after fresh embryo transfers are lighter than those that are NC and that these differences in BW are further associated with differences in child growth up to the age of 5.[39] However, this study was only able to examine the subset of ART children born in Scotland and the dataset created was not readily linkable to other datasets.

## COHORT DESCRIPTION
### Databases used
#### Human Fertilisation and Embryology Authority Register

The HFEA, an 'arm's length body' of the Department of Health, acts as an independent regulator of fertility treatment and research using human embryos in the UK.[31] All licensed fertility clinics in the UK are required by law to provide information to the HFEA about treatments they carry out and their outcomes, ensuring high levels of data completeness. Only 1500 ART outcomes had not been reported as of 07/01/2020, almost all of which were likely to be from overseas patients who had ART in the UK and returned home for the delivery making it difficult for clinics to follow-up.[42] Regular quality assurance checks including manual validation of data submissions; regular quality assurance checks on data through the inspection process; publication of non-compliances with data quality issues in inspection reports and, where relevant, review of quality reports and auditing of clinics with irregular data submissions are also carried out.[43]

The HFEA Act 1990[29] made prospective collection and storage of baseline information and birth outcomes on the HFEA register mandatory, although 'consent for disclosure of information for research' was not collected from patients who underwent treatment at a licensed fertility clinic prior to September 2009. The reliability and completeness of information relating to women who have undergone treatment on the HFEA register are considerably high (approximately 99.9%, 100%–99.9% completeness for forename, surname and DOB, respectively; online supplemental table S1), although it contains little identifiable information on children born after ART (eg, 11.2% and 10.3% completeness for child's surname and forename, respectively; online supplemental table S1).

Details of the data accessed from HFEA for this study are discussed below (see Linkage section).

#### Birth Registration Database

In the UK, all births are legally required to be registered by the Local Registration Service in partnership with the General Register Office (GRO) in England and Wales, making it the most complete data source available.[44] The birth registration dataset is managed by ONS in England and Wales, while Scottish birth records are held by National Records for Scotland (NRS) who work closely with NHS Digital to ensure daily record transfers between the organisations using a secure and closed electronic system.

This study used the birth registration dataset to identify all children born to women known to have undergone ART as well as their NCS, and details of the data accessed have been discussed below (see Linkage section).

#### Hospital Episode Statistics

Hospital Episode Statistics (HES) is a data warehouse containing details of all admissions, outpatient appointments and A&E attendances at NHS hospitals in England.[45]

HES data covers all NHS Clinical Commissioning Groups (CCGs) in England, including private patients treated in NHS hospitals, patients resident outside of England, and care delivered by treatment centres (including those in the independent sector) funded by the NHS.[45] Approximately 98%–99% of hospital activity in England are estimated to be funded by the NHS.[46] Moreover, the HES admitted patient care (APC) database covers all births in NHS hospitals, representing approximately 97.3% of births in England, thus enabling creation of nationally representative birth cohorts.[47 48]

Although the HES data warehouse has been operating since 1990,[49] linkage of the HES APC episodes longitudinally to the same individuals only commenced in 1997/98 when the patient's NHS number became a mandated return from hospitals.[48] Furthermore, linkage of this APC dataset to other HES datasets such as outpatient, A&E and adult critical care only commenced much later in 2003/2004, 2007/2008 and 2008/2009, respectively.[48] HES diagnoses were coded using the International Classification of Diseases version 9 (ICD-9) between April 1989 and March 1995 and ICD-10 thereafter.

This study linked the HES database to the ART children, NCS and matched NCP control groups, and the data accessed for this purpose have been discussed below (see Linkage section). Although the current paper only uses demographic data (ethnicity; UK census-derived Index of Multiple Deprivation (IMD), the official measure of relative deprivation for small areas or neighbourhoods in the UK[50]) and BW from HES to carry out an exemplar analysis, further analyses of longitudinal health outcomes in these cohorts are already underway and will be published shortly.

#### Medical Integrated Database and Administration System

MIDAS is a system used by NHS Digital's Data Linkage and Extract Service (DLES) to replace the functionality previously provided by the Central Health Register Information System (CHRIS) application, specifically for the administration of information used within research and data linkage functions. It was developed to provide continuity for the existing list cleaning, flagging, tracing and cohort management services to maintain support for approved research and audit projects, National Cancer Registration and the ONS Longitudinal Study. It contains up to date demographic GP registration data via a daily feed from the Secondary User Service, standard demographic identifiers such as names and addresses, information on patients who have exited from and returned to the NHS, as well as cancer and death registration details. MIDAS can be considered as a window into the Personal Demographics Service database, providing bespoke data extracts in addition to regular patient tracking services. Extracts are provided on an individual as-and-when basis and are subject to the regular DLES approvals process.

In this study, the cohort produced by Linkage 1 (discussed later) was traced on MIDAS for further demographic information (NHS number where necessary,

postcode, etc) that would allow subsequent linkage to HES. Additionally, the matched NCP controls were also identified using MIDAS.

## Inclusion and exclusion criteria

ART: defined as 'treatments or procedures that include in vitro handling of both human oocytes and sperm or embryos, for the purpose of reproduction'.[51] This study included all children born after non-donor ART conducted in the UK between 1 April 1992 and 31July 2009, in keeping with legislative changes that influenced the ability to link HFEA data to other health datasets.[39 41] It was estimated that only 0.2% of patients receiving ART during the study period retrospectively withdrew consent.[52]

NCS: All NC children born to women with at least one child born following ART were included. This comprised of both full (both parents the same) and maternal half-siblings (same mother different father), but identification of paternal half-siblings was not possible.

NCP controls: The birth registration dataset and MIDAS were used to identify two matched (by month and year of birth, sex and multiplicity/plurality) NCP controls per ART child (ART: control=1:2).

All children conceived in the UK after ART but born outside of England, Wales and Scotland, those born after ART to women who permanently lived outside the UK but travelled to the UK for ART treatment and those born in Northern Ireland were excluded as it would not be possible to link them to ONS birth records. Additionally, siblings born outside of the study period (as their conception status could not be verified) as well as those born outside of England, Wales and Scotland were also excluded. Cases that had withdrawn consent for their data to be used for research and children born after donor ART were excluded in compliance with HFEA legislation that prevents the viewing of identifiable data relating to these children by any third party.

## Linkage

As described above, the HFEA Register contains very few identifiers for children born after ART, thus limiting linkage to other nationally held datasets. Therefore, this study initially linked HFEA records of children born to women known to have undergone ART to nationally held birth records to add identifiers and NHS numbers, which would enable further linkage with a variety of other datasets (shown in figures 1 and 2).

### Linkage 1: HFEA-ONS

Step 1 (1 and 2 from figure 1): The HFEA first identified children born after ART and assigned them pseudoanonymised unique record numbers (URN). Maternal unique record numbers (mURN; created from a previous study[52] online supplemental table S2) were added to the file, which was encrypted and sent securely to NHS Digital.

This file contained the following variables: date of birth (DOB) of child, BW of child, sex of child, forename of mother, surname of mother, previous name(s) of mother, multiplicity/plurality (eg, singleton, twin), and mother's DOB.

Step 2 (3 and 4 from figure 1): NHS Digital created file 2 (from a previous study[52]; online supplemental table S2) containing extracts of women who had undergone ART within the specified time period and securely transferred it to ONS. This file contained the NHS Number, DOB of possible mother, surname of possible mother, forename of possible mother, other name(s) of possible mother, and the HFEA mURN.

ONS then used these variables to link to birth records to create an extract of all children born to these mothers within the specified time frame. This extract (File 3) contained the mothers' details as supplied in File 2, surname of the child, forename of the child, DOB of the child, NHS number of the child (if available), BW of the child and sex of the child. This file was then securely transferred back to NHS Digital.

Step 3 (5, 6, 7, 8 and 9 from figure 1): NHS Digital then matched files 1 and 3 to determine children born to these mothers after ART and their NCS. This linkage was carried out using DOB of child, NHS number of child (if available), BW of child, sex of child, surname of child (where available on HFEA file 1) and forename of child (where available on HFEA file 1).

The ART and sibling cohorts were then traced using NHS Digital's national Medical Integrated Database and Administration System (MIDAS) to update demographic information (including NHS number, postcode) and current status (emigrations/deaths, etc) and allow subsequent linkage to medical records.

Step 4 (10 and 11 from figure 1): NHS Digital used MIDAS to identify two controls for every cohort member, matched for month and year of birth, sex and multiplicity/plurality (singleton/ multiple birth).

Step 5 (14 from figure 1): NHS Digital produced updated demographic information on all ART children, siblings and controls (as obtained from MIDAS) and provided de-identified outputs plus non-identifiable deprivation scores to the research team at University College London (UCL).

### Linkage 2: HFEA-ONS-HES

Step 6 (15 on figure 2): NHS Digital then linked all cohorts produced by linkage 1 to HES using NHS number, postcode and DOB and shared with the research team at UCL.

Step 7 (16 on figure 2): HFEA staff extracted relevant anonymised fertility data and this was shared with the research team at UCL who linked these two files (from steps 6 and 7) using the URNs.

Step 8 (17 on figure 2): For the purpose of this second linkage, UCL researchers excluded records of children born before 1 April 1997 to coincide with the start of HES monitoring as well as triplets and higher order births from all groups (along with associated ART and NC births) to produce the final cohorts for health outcome analysis.

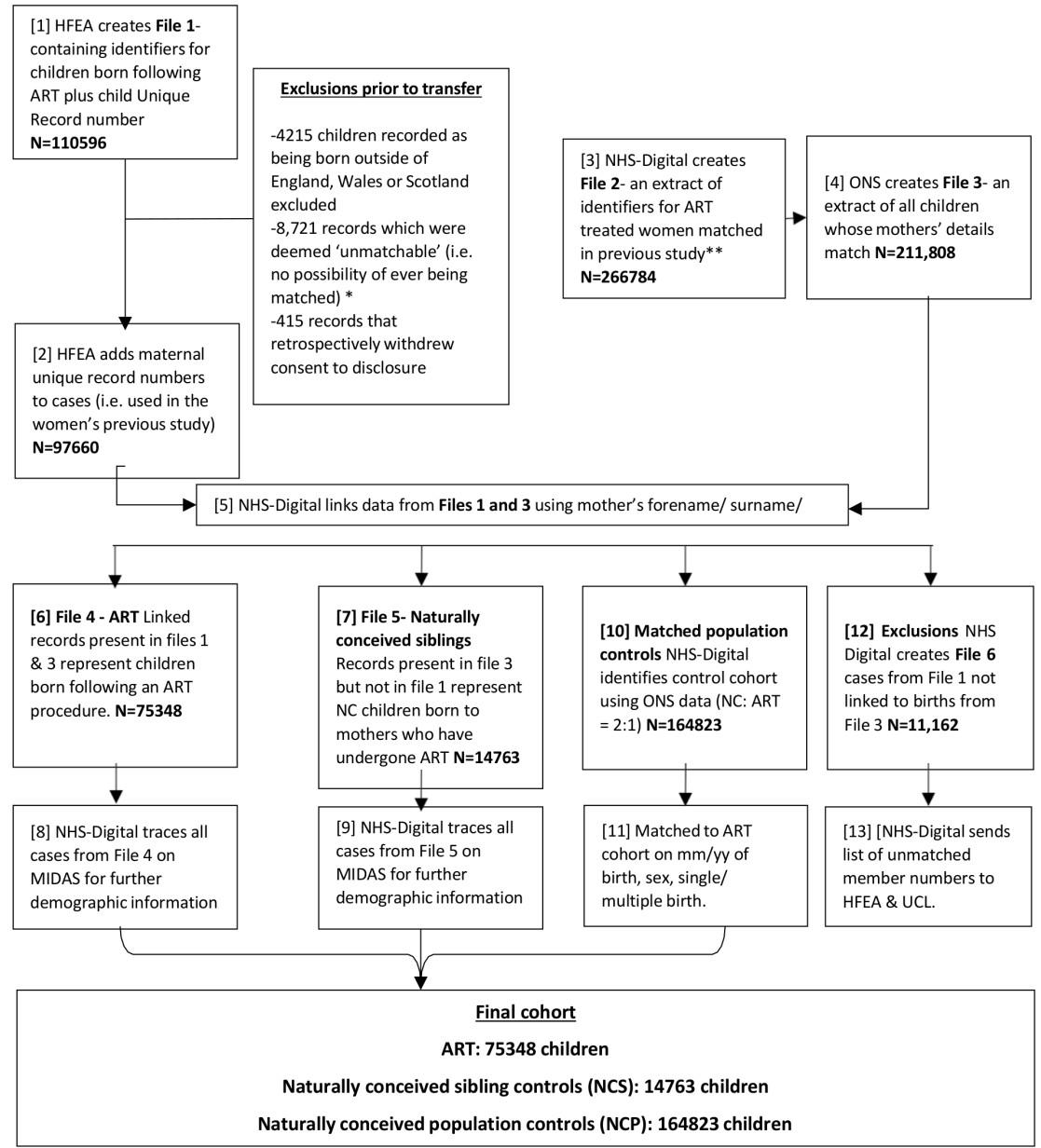

*as they were a) births outside of England/ Wales; b) births before 1993 (when ONS systems were automated and thus the date
from which linkage is possible to ONS records); and c) to mothers which were not included in file 2 (as it was not possible to
identify them on NHS-Digital systems previously- 'women's study').
** Please see Supplementary figure S2 for cohort flow

**Figure 1** Flowchart showing linkage 1 (HFEA-ONS). *As they were (a) births outside of England/ Wales; (b) births before 1993 (when ONS systems were automated and thus the date from which linkage is possible to ONS records) and (c) to mothers which were not included in file 2 (as it was not possible to identify them on NHSDigital systems previously—'women's study'). **Please see online supplemental figure S2 for cohort flow. ART, assisted reproductive technology; HFEA, Human Fertilisation Embryology Authority; NC, naturally conceived; ONS, Office for National Statistics.

Step 9 (18 on figure 2): NHS Digital produced file 4 (shown in figure 1) for the ART and NC cohorts and these are now securely stored at HFEA and NHS Digital, respectively.

The entire linkage process was completed by NHS Digital and none of the researchers had access to any identifiable participant data. The details of the ART and NC groups were encrypted and are currently held securely at HFEA and NHS Digital, respectively.

## Cohorts produced
### Linkage 1: HFEA-ONS cohort

1. ART: This group consisted of all children born after non-donor ART conducted in the UK between 1 April 1992 and 31st July 2009.
2. NCS: This included all NC children born to women who had at least one child born following ART. This included full and maternal half-siblings but not paternal half-siblings.

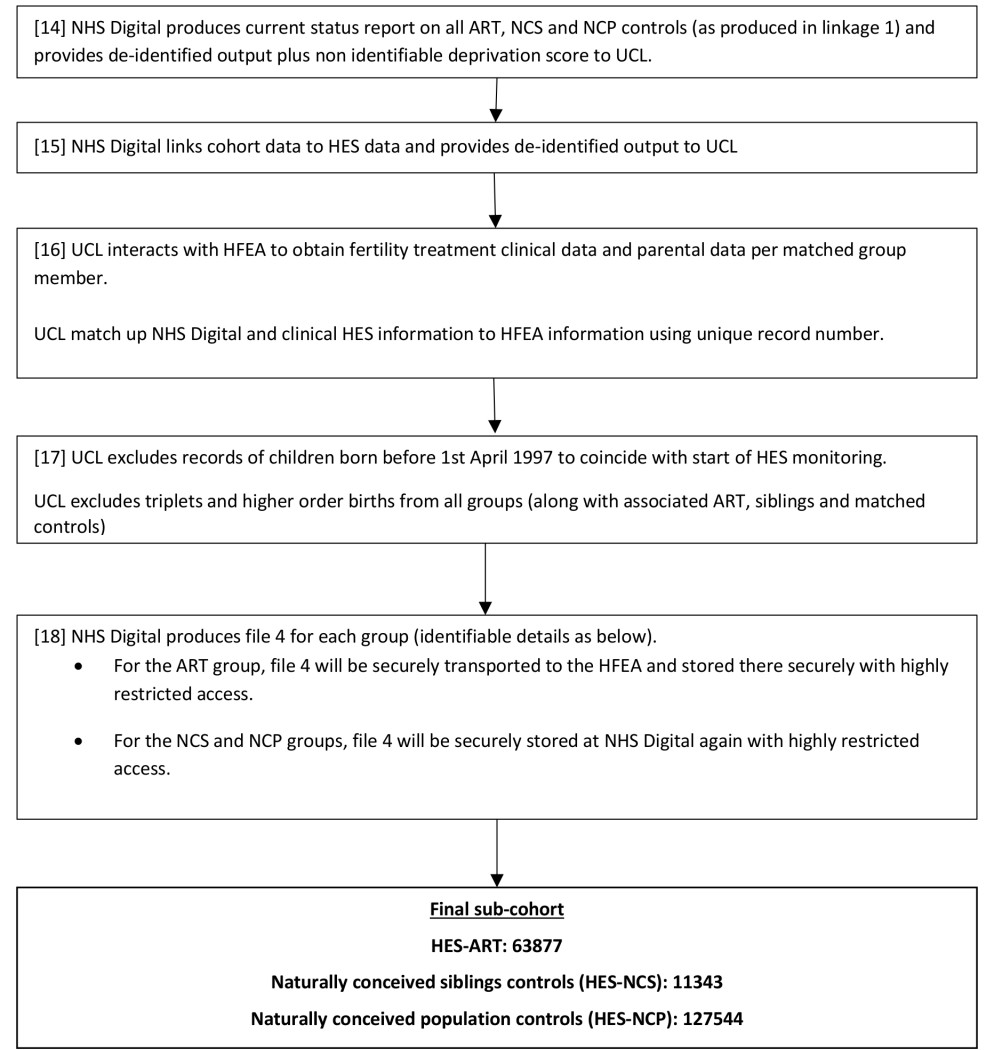

[14] NHS Digital produces current status report on all ART, NCS and NCP controls (as produced in linkage 1) and provides de-identified output plus non identifiable deprivation score to UCL.

[15] NHS Digital links cohort data to HES data and provides de-identified output to UCL

[16] UCL interacts with HFEA to obtain fertility treatment clinical data and parental data per matched group member.

UCL match up NHS Digital and clinical HES information to HFEA information using unique record number.

[17] UCL excludes records of children born before 1st April 1997 to coincide with start of HES monitoring.

UCL excludes triplets and higher order births from all groups (along with associated ART, siblings and matched controls)

[18] NHS Digital produces file 4 for each group (identifiable details as below).
- For the ART group, file 4 will be securely transported to the HFEA and stored there securely with highly restricted access.
- For the NCS and NCP groups, file 4 will be securely stored at NHS Digital again with highly restricted access.

**Final sub-cohort**
**HES-ART: 63877**
**Naturally conceived siblings controls (HES-NCS): 11343**
**Naturally conceived population controls (HES-NCP): 127544**

**Figure 2** Flowchart showing linkage 2 (HFEA-ONS-HES). ART, assisted reproductive technology; HES, Hospital Episode Statistics database; HFEA, Human Fertilisation and Embryology Authority; UCL, University College London.

3. NCP: This consisted of two matched (by month and year of birth, sex and multiplicity/plurality) NCP controls for each ART child.

### Linkage 2: HFEA-ONS-HES subcohort
1. ART with HES outcomes (HES-ART): This included all children born after non-donor ART conducted in the UK between 1 April 1997 and 31 July 2009 linked to their HES outcomes.
2. NCS with HES outcomes (HES-NCS): This included the NCS of all non-donor ART children born in the UK between 1 April 1997 and 31 July 2009 linked to their HES outcomes.
3. NCP control with HES outcomes (HES-NCP): This included two matched NCP controls for each non-donor ART child born in the UK between 1 April 1997 and 31 July 2009 linked to their HES outcomes.

### Data cleaning
Data cleaning included deletion of triplets and higher order births along with their NCS and matched NCP controls; deletion of duplicates; and reformatting,

labelling and creation of new variables. Triplets and higher order births were excluded from analysis as they are known to be associated with adverse perinatal outcomes such as higher infant mortality, birth defects, premature birth and low BW.[53 54] Initial data scoping also revealed several discrepancies in the data, a common limitation of large data linkage studies utilising multiple data sources, and pragmatic rules were employed to derive a consensus. Additional data cleaning carried out on the subcohort produced via linkage 2 included creation of an additional group consisting of ART children who had NCS (HES-sART) for the purposes of analysis. Order of pregnancy was calculated by sorting the child DOBs within each family in ascending order (online supplemental table S3), and the type of ART and cause of infertility variables were regrouped appropriately.

### Exemplar analysis
An exemplar analysis examining the effect of fresh and frozen embryo transfer (each compared with NC) on singleton BW was carried out using the HFEA-ONS

cohort and HFEA-ONS-HES subcohort (inclusion criteria discussed previously). Details of BW are provided by the hospital where the birth took place or by the midwife/doctor in attendance at the birth and is then passed to ONS as a consequence of the NHS birth notification being linked to the corresponding birth registration by the registrar. Multiple regression analysis was used to compare BW between children born after ART and NCP controls. The models were adjusted for maternal age at delivery and sex for the ART versus NCP analysis and for maternal age at delivery, sex, IMD at earliest appointment and ethnicity for the HES-ART vs HES-NCP analysis.

A family-matched model was used to compare BW between ART children and their NCS, including a family covariate to allow for within family correlations. The models were adjusted for maternal age at delivery, sex and order of pregnancy. IMD and ethnicity were excluded from this analysis as the underlying effects they represent would have remained constant within families. The models were parameterised to directly estimate the differences between ART (fresh and frozen) and NC children. The fresh versus frozen difference was estimated from the model by construction of the appropriate contrast. All analyses were carried out using Stata V.16.0.

This paper was developed in accordance with the STROBE reporting guidelines.[55]

### Patient and public involvement

No patients involved. Due to the very personal nature of the treatments involved, it was not appropriate to contact the families directly, thus preventing us from involving patients or the public in the design, conduct, reporting, or dissemination plans of our research.

## FINDINGS TO DATE
### Final HFEA-ONS cohort

The HFEA identified 110 596 children that met our inclusion criteria, of which 97 660 (figure 1) were deemed suitable for transfer to NHS Digital and subsequent linkage to ONS birth records. The linkage success rate was 77% (75 348 cases out of 97 660) due to the poor quality of data received from the HFEA.

A feasibility pilot study testing the validity of the linkage carried out in this study showed that the false positivity rates were very low (<0.0002%, unpublished data). Matching was done using SQL using exact matching (design to be inclusive of all potential matches) followed by probabilistic matching using Jaro Winkler software. A clash was defined as a complete incompatibility of information between databases for a particular variable, with no clashes deemed as being a match. The HFEA were asked for further information in case of one clash, and more than one clash was considered to be a failed match. In the current study, the main linkage was carried out using a very high threshold for matching, and the weak/inaccurate identifiers in the HFEA register often resulted in matches that failed to pass this threshold. The majority of unlinked records could not be found due to clashes in parental identifiers and child date of birth.

As the main linkage carried out in this study was bespoke in nature, particularly with regard to identification of the NC siblings, and had not been carried out before, standardised quality assurance measures could not be applied and instead extensive manual checking of the linkage was implemented. However, due to the mandatory nature of reporting all ART cycles carried out in the UK to the HFEA, it is unlikely that the unlinked records differed significantly from those that were linked, thus minimising the risk of selection bias.

Tables 1 and 2 show the descriptive characteristics of the final HFEA-ONS cohort generated after completion of Linkage 1 (figure 1) and data cleaning. This cohort included 75 348 children born after ART between 1992 and 2009 and 164 823 matched NCP controls. Of the children born after ART, 15 875 (sART) had 14 763 NCS and the majority of families had had an ART pregnancy first followed by natural conception (online supplemental table S3).

The majority of the final cohort were ethnically white and singleton births, and a greater proportion of the ART and NCS groups were from a higher socioeconomic class. The majority of women who underwent ART were aged between 30 and 39 years at the time of delivery, while those with NC births tended to be slightly younger (25–39 years).

### Final HFEA-ONS-HES sub-cohort

All births before 1 April 1997 were excluded for the purpose of linkage to HES, resulting in 63877, 11 343 and 127 544 children in the HES-ART, HES-NCS and HES-NCP groups, respectively. Linkage 2 (figure 2) had a success rate of 84.7%, and the descriptive characteristics of the sub-cohort produced have been shown in tables 2 and 3

### Exemplar analysis: singleton BW

The exemplar analysis included all singleton births in the HFEA-ONS cohort and HFEA-ONS-HES subcohort.

HFEA-ONS: Comparing ART to NCP showed that children born after fresh embryo transfers were lighter (BW difference: −131 g; 95% CI: −140 to −123) and those born after frozen embryo transfers were heavier (BW difference: 35 g; 95% CI: 19 to 52) than the NC controls (table 4; online supplemental table S4). Moreover, children born after frozen embryo transfers were significantly heavier than those born after fresh embryo transfers (BW difference: −167 g; 95% CI: −183 to −150).

Family-matched analysis showed that children born after fresh embryo transfers were lighter (BW difference: −54 g; 95% CI: −72 to −36) and those born after frozen embryo transfers (BW difference: 138 g; 95% CI: 101 to 175) were heavier than the NCS group (table 4; online supplemental table S5). Moreover, children born after frozen embryo transfers were significantly heavier than

**Table 1** Demographic characteristics of art children, NCS and matched NCP controls born between 1 April 1992 and 31 July 2009

| | HFEA-ONS cohort | | | |
| --- | --- | --- | --- | --- |
| | **ART** | **Control (NCP)** | **ART with siblings (sART)** | **Siblings (NCS)** |
| **Infants** | 75 348 | 164 823 | 15 875 | 14 763 |
| Mean BW (SD) | | | | |
| Singleton | 3282.0 (620.3) | 3353.83 (578.5) | 3311.12 (605.1) | 3430.05 (581.9) |
| Multiple | 2378.3 (593.4) | 2340.28 (626.1) | 2405.10 (608.1) (638.36) | 2292.03 (780.5) |
| Sex | | | | |
| Female | 36 994 (49.1%) | 80 859 (49.1%) | 7598 (47.9%) | 7271 (49.3%) |
| Male | 38 354 (50.9%) | 83 964 (50.9%) | 8277 (52.1%) | 7492 (50.8%) |
| Multiplicity/plurality | | | | |
| Singleton | 44 488 (59.1%) | 99 554 (60.4%) | 10 592 (66.7%) | 14 055 (95.2%) |
| Twin | 30 860 (40.9%) | 65 269 (39.6%) | 5283 (33.3%) | 708 (4.8%) |
| Maternal age group (years) | | | | |
| <25 | 731 (0.1%) | 33 156 (20.1%) | 256 (1.6%) | 388 (2.6%) |
| 25–29 | 5315 (7.1%) | 30 231 (18.3%) | 1326 (8.4%) | 1032 (6.1%) |
| 30–34 | 23 342 (30.1%) | 46 050 (27.9%) | 5466 (34.4%) | 3855 (26.1%) |
| 35–39 | 29 187 (38.7%) | 30 345 (18.4%) | 5872 (36.1%) | 5711 (38.7%) |
| 40–44 | 8705 (11.6%) | 7812 (4.7%) | 1315 (8.3%) | 2157 (14.6%) |
| ≥45 | 84 (0.1 %) | 587 (0.4%) | 23 (0.1%) | 53 (0.4%) |
| Missing | 7984 (10.6%) | 16 642 (10.1%) | 1617 (10.2%) | 1567 (10.6%) |
| Birth year | | | | |
| 1992 | 57 (0.1%) | 131 (0.1%) | 12 (0.08%) | 23 (0.2%) |
| 1993 | 1595 (2.1%) | 3187 (1.9%) | 299 (1.8%) | 337 (2.3%) |
| 1994 | 2021 (2.7%) | 4029 (2.4%) | 425 (2.7%) | 389 (2.6%) |
| 1995 | 2439 (3.2%) | 4911 (2.1%) | 518 (3.3%) | 526 (3.6%) |
| 1996 | 2977 (3.1%) | 5989 (3.6%) | 641 (4.0%) | 577 (3.9%) |
| 1997 | 3574 (4.7%) | 7172 (4.4%) | 776 (4.9%) | 679 (4.6%) |
| 1998 | 3826 (5.1%) | 7670 (4.7%) | 861 (5.4%) | 800 (5.4%) |
| 1999 | 4253 (5.6%) | 8531 (5.2%) | 1023 (6.4%) | 804 (5.5%) |
| 2000 | 4504 (5.1%) | 9106 (5.5%) | 1119 (7.1%) | 961 (6.5%) |
| 2001 | 4754 (6.3%) | 9677 (5.9%) | 1163 (7.3%) | 991 (6.7%) |
| 2002 | 5149 (6.8%) | 10 464 (6.4%) | 1339 (8.4%) | 999 (6.8%) |
| 2003 | 5531 (7.3%) | 11 373 (6.9%) | 1401 (8.8%) | 1064 (7.2%) |
| 2004 | 5671 (7.5%) | 11 835 (7.2%) | 1379 (8.7%) | 1098 (7.4%) |
| 2005 | 5768 (7.7%) | 12 310 (7.5%) | 1372 (8.6%) | 1109 (7.5%) |
| 2006 | 6331 (8.4%) | 14 248 (8.6%) | 1273 (8.0%) | 1116 (7.6%) |
| 2007 | 6410 (8.5%) | 15 976 (9.7%) | 1126 (7.1%) | 1209 (8.9%) |
| 2008 | 6395 (8.5%) | 17 222 (10.5%) | 719 (4.5%) | 1271 (8.6%) |
| 2009 | 4092 (5.4%) | 10 992 (6.7%) | 429 (2.7%) | 810 (5.5%) |

ART, assisted reproductive technology; BW, birth weight; HFEA, Human Fertilisation and Embryology Authority; NCP, naturally conceived population controls; NCS, naturally conceived siblings; ONS, Office for National Statistics; sART, ART with siblings.

those born after fresh embryo transfers (BW difference =−193g; 95% CI: −232 to −154).

HFEA-ONS-HES: Comparing HES-ART to HES-NCP showed that children born after fresh embryo transfers were lighter (BW difference: −152 g; 95% CI: −162 to −142) and those born after frozen embryo transfers were similar in weight (BW difference: 3 g; 95% CI: −17 to 22) to the NC controls (table 4; online supplemental table

**Table 2** Fertility characteristics of ART children by cohort

| | HFEA-ONS cohort | | HFEA-ONS-HES subcohort | |
|---|---|---|---|---|
| | **ART** | **sART (ART with siblings)** | **HES-ART** | **HES-sART (ART with siblings)** |
| Multiplicity/plurality | | | | |
| Singletons | 44 489 (59.0%) | 10 592 (66.7%) | 37 891 (59.4%) | 8383 (67.1%) |
| Twins | 30 859 (40.1%) | 5283 (33.3%) | 25 986 (40.6%) | 3946 (32.0%) |
| Paternal age at childbirth | | | | |
| <25 | 406 (0.5%) | 122 (0.8%) | 324 (0.5%) | 87 (0.7%) |
| 25–29 | 4878 (6.5%) | 1206 (7.6%) | 3911 (6.1%) | 938 (7.6%) |
| 30–34 | 21 526 (28.6%) | 5304 (33.4%) | 17 706 (27.7%) | 4116 (33.4%) |
| 35–39 | 27 678 (36.7%) | 6007 (37.8%) | 23 626 (36.1%) | 4692 (38.1%) |
| 40–44 | 13 458 (17.9%) | 2306 (14.5%) | 11 715 (18.3%) | 1778 (14.4%) |
| 45–49 | 4551 (6.0%) | 604 (3.8%) | 4010 (6.3%) | 468 (3.8%) |
| 50–54 | 1636 (2.2%) | 190 (1.2%) | 1440 (2.3%) | 146 (1.2%) |
| ≥55 | 848 (1.1%) | 82 (0.5%) | 787 (1.2%) | 65 (0.5%) |
| Missing | 367 (0.5%) | 54 (0.4%) | 348 (0.6%) | 39 (0.3%) |
| Type of ART | | | | |
| ICSI | 26 629 (38.6%) | 4409 (30.1%) | 26 210 (44.6%) | 3995 (34.1%) |
| IVF | 42 096 (61.0%) | 10 170 (69.4%) | 32 282 (54.1%) | 7364 (64.4%) |
| IVF:ICSI | 248 (0.4%) | 73 (0.5%) | 247 (0.4%) | 72 (0.6%) |
| Cause of infertility | | | | |
| Male factor | 29 203 (38.8%) | 5675 (35.8%) | 24 876 (38.1%) | 4198 (34.1%) |
| Endometriosis | 2225 (2.1%) | 495 (3.1%) | 1962 (3.1%) | 401 (3.3%) |
| Ovulatory | 8581 (11.4%) | 2198 (13.9%) | 7713 (12.1%) | 1854 (15.1%) |
| Tubal | 19 915 (26.4%) | 3450 (21.7%) | 15 041 (23.6%) | 2378 (19.3%) |
| Unknown cause | 15 387 (20.4%) | 4048 (25.5%) | 14 252 (22.3%) | 3491 (28.3%) |
| Fresh/frozen transfer | | | | |
| Fresh | 66 534 (88.3%) | 14 079 (88.7%) | 56 230 (88.0%) | 10 907 (88.5%) |
| Frozen | 8739 (11.6%) | 1782 (11.2%) | 7595 (11.9%) | 1414 (11.7%) |
| Missing | 75 (0.1%) | 14 (0.1%) | 52 (0.1%) | 8 (0.1%) |
| Previous live births | | | | |
| 0 | 67 235 (89.2%) | 14 878 (93.7%) | 56 250 (88.1%) | 11 498 (93.3%) |
| 1 | 7829 (10.4%) | 958 (6.1%) | 7348 (11.5%) | 795 (6.5%) |
| 2 | 273 (0.4%) | 35 (0.2%) | 268 (0.4%) | 32 (0.3%) |
| 3 | 9 (0.0%) | 2 (0.0%) | 9 (0.1%) | 2 (0.0%) |
| Missing | 2 (0.0%) | 2 (0.0%) | 2 (0.0%) | 2 (0.0%) |
| Parity of mother (previous pregnancies as recorded on the register) | | | | |
| 0 | 64 502 (85.6%) | 14 270 (89.9%) | 53 987 (84.5%) | 11 029 (89.5%) |
| 1 | 9798 (13.0%) | 1450 (9.2%) | 8901 (13.9%) | 1167 (9.5%) |
| 2 | 949 (1.3%) | 140 (0.9%) | 894 (1.4%) | 119 (0.1%) |
| 3 | 83 (0.1%) | 9 (0.1%) | 79 (0.1%) | 8 (0.1%) |
| 4 | 14 (0.0%) | 4 (0.0%) | 14 (0.0%) | 4 (0.0%) |
| Null | 2 (0.0%) | 2 (0.0%) | 2 (0.0%) | 2 (0.0%) |
| Infertility duration in years | | | | |
| Mean (SD) | 4.78 (2.9) | 4.37 (2.6) | 4.62 (2.8) | 4.31 (2.6) |
| Missing | 17 451 (23.2%) | 4951 (31.2%) | 17 898 (24.9%) | 5364 (27.1%) |

ART, assisted reproductive technology; HES, Hospital Episode Statistics; HFEA, Human Fertilisation and Embryology Authority; ICSI, intracytoplasmic sperm injection; IVF, in vitro fertilisation; ONS, Office for National Statistics; sART, ART with siblings.

**Table 3** Demographic characteristics of ART children, NCS and matched NCP controls born between 1 April 1997 and 31 July 2009

| | HFEA-ONS-HES subcohort | | | |
| --- | --- | --- | --- | --- |
| | **HES-ART** | **Control (HES-NCP)** | **ART with siblings (HES-sART)** | **Siblings (HES-NCS)** |
| Infants | 63 877 | 127 544 | 12 329 | 11 343 |
| Follow-up period in days | | | | |
| Median (IQR) | 4429 (2181) | 4409 (2141) | 4635 (1846) | 4307 (2029) |
| Mean BW (SD) | | | | |
| Singleton | 3166.95 (742.3) | 3271.67 (648.5) | 3222.25 (699.3) | 3346.54 (712.0) |
| Multiple | 2172.27 (715.4) | 2155.77 (683.9) | 2201.58 (724.9) | 2301.17 (677.8) |
| Sex | | | | |
| Female | 31 435 (49.2%) | 62 785 (49.2%) | 5907 (47.9%) | 5573 (49.1%) |
| Male | 32 442 (50.8%) | 64 759 (50.8%) | 6422 (52.1%) | 5770 (50.9%) |
| Multiplicity/plurality | | | | |
| Singleton | 37 890 (59.4%) | 75 642 (59.3%) | 8383 (68.0%) | 10 815 (95.8%) |
| Twin | 25 987 (40.6%) | 51 902 (40.7%) | 3946 (31.1%) | 528 (4.2%) |
| IMD decile at earliest appointment | | | | |
| 1 (most deprived) | 2045 (3.2%) | 12 696 (9.9%) | 349 (2.8%) | 299 (2.6%) |
| 2 | 2650 (4.2%) | 11 104 (8.7%) | 458 (3.7%) | 396 (3.5%) |
| 3 | 3311 (5.2%) | 10 057 (7.8%) | 539 (4.4%) | 463 (4.1%) |
| 4 | 3926 (6.1%) | 9550 (7.5%) | 678 (5.4%) | 588 (5.2%) |
| 5 | 4666 (7.2%) | 9161 (7.5%) | 862 (6.1%) | 746 (6.6%) |
| 6 | 5350 (8.4%) | 8872 (6.5%) | 1084 (8.8%) | 921 (8.1%) |
| 7 | 6142 (9.6%) | 8842 (6.9%) | 1205 (9.8%) | 1044 (9.2%) |
| 8 | 6675 (10.5%) | 8925 (7.0%) | 1299 (10.5%) | 1161 (10.3%) |
| 9 | 7729 (12.1%) | 9083 (7.1%) | 1557 (12.6%) | 1386 (12.2%) |
| 10 (least deprived) | 7710 (12.1%) | 8299 (6.5%) | 1720 (14.0%) | 1499 (13.3%) |
| Missing | 13 673 (21.4%) | 30 955 (24.3%) | 2580 (20.9%) | 2840 (25.0%) |
| Birth year | | | | |
| 1997 | 2597 (4.1%) | 5160 (4.1%) | 518 (4.2%) | 242 (2.1%) |
| 1998 | 3708 (5.8%) | 7389 (5.9%) | 743 (6.0%) | 419 (3.7%) |
| 1999 | 4083 (6.4%) | 8126 (6.5%) | 852 (6.9%) | 560 (4.9%) |
| 2000 | 4310 (6.8%) | 8633 (6.9%) | 925 (7.5%) | 781 (6.9%) |
| 2001 | 4559 (7.1%) | 9160 (7.3%) | 972 (7.9%) | 879 (7.78%) |
| 2002 | 4980 (7.8%) | 9933 (7.9%) | 1171 (9.5%) | 937 (8.3%) |
| 2003 | 5379 (8.4%) | 10 788 (8.5%) | 1253 (10.2%) | 1012 (8.9%) |
| 2004 | 5561 (8.7%) | 11 082 (8.7%) | 1271 (10.3%) | 1067 (9.4%) |
| 2005 | 5662 (8.9%) | 11 326 (8.8%) | 1271 (10.3%) | 1078 (9.5%) |
| 2006 | 6275 (9.8%) | 12 513 (9.7%) | 1217 (9.7%) | 1100 (9.7%) |
| 2007 | 6342 (9.9%) | 12 701 (9.8%) | 1058 (8.6%) | 1199 (10.6%) |
| 2008 | 6347 (9.9%) | 12 718 (9.8%) | 670 (5.4%) | 1260 (11.1%) |
| 2009 | 4074 (6.4%) | 8015 (6.2%) | 408 (3.3%) | 809 (7.1%) |
| Ethnicity | | | | |
| White | 61 921 (96.9%) | 122 050 (95.7%) | 11 983 (97.2%) | 11 084 (97.7%) |
| Asian/Asian British | 959 (1.5%) | 2496 (1.1%) | 197 (1.6%) | 153 (1.4%) |
| Chinese | 35 (0.1%) | 89 (0.1%) | 8 (0.1%) | 1 (0.0%) |

Continued

**Table 3** Continued

| | HFEA-ONS-HES subcohort | | | |
|---|---|---|---|---|
| | **HES-ART** | **Control (HES-NCP)** | **ART with siblings (HES-sART)** | **Siblings (HES-NCS)** |
| Black/African/Caribbean/Black British | 433 (0.7%) | 5268 (4.0%) | 61 (0.5%) | 39 (0.3%) |
| Mixed/multiple ethnic groups | 318 (0.5%) | 721 (0.6%) | 38 (0.3%) | 27 (0.2%) |
| Other ethnic group | 211 (0.3%) | 498 (0.4%) | 42 (0.3%) | 39 (0.3%) |
| Maternal age at delivery | | | | |
| ≤25 | 710 (1.1%) | 27 783 (21.9%) | 241 (1.1%) | 317 (2.9%) |
| 25–29 | 5085 (7.1%) | 25 115 (19.7%) | 1209 (9.9%) | 887 (7.8%) |
| 30–34 | 21 994 (34.4%) | 38 896 (30.5%) | 4797 (38.9%) | 3309 (29.1%) |
| 35–39 | 27 682 (43.4%) | 25 907 (20.3%) | 4998 (40.6%) | 4954 (43.7%) |
| 40–44 | 8164 (12.1%) | 6419 (5.2%) | 1057 (8.6%) | 1830 (16.2%) |
| ≥45 | 217 (0.3%) | 772 (0.6%) | 20 (0.2%) | 45 (0.4%) |
| Missing | 25 (0.0%) | 2652 (2.0%) | 7 (0.0%) | 1 (0.0%) |

ART, assisted reproductive technology; BW, birth weight; HES, Hospital Episode Statistics; HFEA, Human Fertilisation and Embryology Authority; ICSI, intracytoplasmic sperm injection; IMD, Index of Multiple Deprivation; IVF, in vitro fertilisation; NCP, Naturally conceived population control; NCS, Naturally conceived siblings; ONS, Office for National Statistics; sART, ART with siblings.

S6). Moreover, children born after frozen embryo transfers were significantly heavier than those born after fresh embryo transfers (BW difference:−155 g; 95% CI: −175 to −135).

Family-matched analysis comparing showed that children born after fresh embryo transfers were lighter (BW difference: −57 g; 95% CI: −75 to −38) and those born after frozen embryo transfers were heavier (BW difference: 152 g; 95% CI: 113 to 190) than the HES-NCS group (table 4; online supplemental table S7). Moreover, children born after frozen embryo transfers were significantly heavier than those born after fresh

embryo transfers (BW difference: −209 g; 95% CI: −249 to −168).

## DISCUSSION

This study aimed to substantially enhance the research value of the HFEA register by utilising electronic record linkage methodology to establish a cohort of ART children born in the UK between 1992 and 2009, their NCS and matched NCP controls. Additionally, a subcohort consisting of those born between 1997 and 2009 was also linked to the Hospital Episode Statistics database

**Table 4** Statistical analysis of BW

| | HFEA-ONS cohort *BW coefficient (95% CI)* | | HFEA-ONS-HES subcohort *BW coefficient (95% CI)* | |
|---|---|---|---|---|
| | **ART (N=75 348)—NCP (N=1 64 823) *adjusted for maternal age at delivery and sex** | **sART (N=15 875)—NCS (N=14 763) *adjusted for maternal age at delivery, sex and order of pregnancy** | **HES-ART (N=63 877)—HES-NCP (N=1 27 544) *adjusted for maternal age at delivery, sex, IMD at earliest appointment and ethnicity** | **HES-sART (N=12 329)—HES-NCS (N=11 343) *adjusted for maternal age at delivery, sex and order of pregnancy** |
| Fresh vs NC (g) | −131 (−140 to 123) | −54 (−72 to 36) | −152 (−162 to 142) | −57 (−75 to −38) |
| Frozen vs NC (g) | 35 (19 to 52) | 138 (101 to 175) | 3 (−17 to 22) | 152 (113 to 190) |
| Fresh vs frozen (g) | −167 (−183 to 150) | −193 (−232 to 154) | −155 (−175 to 135) | −209 (−249 to −168) |

ART, assisted reproductive technology; BW, birth weight; HES, Hospital Episode Statistics; HFEA, Human Fertilisation and Embryology Authority; IMD, Index of Multiple Deprivation; NC, naturally conceived; NCP, Naturally conceived population control; NCS, Naturally conceived siblings; ONS, Office for National Statistics; sART, ART with siblings.

to allow examination of postnatal health outcomes. The final cohort consisted of 75 348 children born after non-donor ART carried out in the UK between 1 April 1992 and 31 July 2009, 14 763 NCS and 164 823 matched NCP controls. Of these, 63 877 ART, 11 343 NCS and 127 544 matched NCP controls were linked to hospital data up to 2015, thus providing a valuable resource for comprehensive, non-invasive, continued immediate and longer term health monitoring of this population as they grow up.

An exemplar analysis comparing BW between children born after ART, their NCS and matched NCP controls was also carried out to demonstrate the validity of this cohort. The results of this analysis confirmed the findings of our previous linkage study on a subset of the entire 1992–2009 cohort,[39] that children born after fresh embryo transfer tended to be lighter and those born from frozen transfer were heavier compared with their NCS and matched NCP controls. These findings are broadly in agreement with numerous large register-based studies and meta-analyses, but add valuable confirmation of the magnitude of BW differences in a large UK population.[15 56–64] Castillo et al[65] reported a similar magnitude of difference to that seen in sibling pairs in a more recent UK cohort, suggesting that the difference in BW between the fresh and frozen embryo transfer groups may not be explained entirely by maternal characteristics and the underlying cause of infertility as these were likely to have remained relatively stable between siblings. Notably, the BW difference observed between frozen transfer and NC babies is much larger in the sibling pair analysis than in the non-sibling comparison and larger than observed by Hann et al.[39] This merits further investigation, especially as increased rates of macrosomia following frozen transfer are an ongoing concern.[34 61] Other possible explanations include differences in the maternal uterine environment caused by either hormonal stimulation (fresh cycles) or hormonal preparation of the uterus (frozen cycles).[66] Although lack of appropriate data on the HFEA register prevented exploration of these effects in the current analysis, a recent UK cohort study by Castillo et al[65] using more detailed clinic data reported no association between type of FET and BW. Molecular changes in the early embryo during freezing and thawing might also result in changes in the developmental process leading to altered BW and long-term health.[33 67–70]

In the current study, the difference in BW between children born after fresh embryo transfer and those born after natural conception was seen to attenuate markedly from −131 g in the population to −54 g in the within-sibling analysis. By contrast, the difference in BW between children born after frozen embryo transfer and those born after natural conception was seen to markedly increase from 35 g in the population to 138 g in the within-sibling analysis in the same dataset. This suggests that the findings of the within-family analysis must be interpreted with caution, with potential explanatory factors for these differences including subfertility and familial masking confounders in relation to the increase of the association of frozen

transfer versus natural conception. Unavailability of birth order data for the population controls prevented us from adjusting for it in the ART versus NCP comparison, and this could plausibly act as a strong confounder given its association with mean BW is stronger than any differences due to ART in most studies. This notable difference between the population and within-family analyses requires further exploration.

Repeating the exemplar analysis using the cohort and subcohort produced by the linkages described above not only increased confidence in the findings but also allowed us to control for potential confounding factors such as maternal age, birth order, socioeconomic status and ethnicity which are known to affect BW.[71]

The cohort described here will allow more effective use of HFEA data for health monitoring and will be of value to researchers from a variety of professional backgrounds. The linked dataset will be returned to the HFEA and access will be controlled by them, although specific ethical approval will be required along with that of the Confidentiality Advisory Group (CAG) of the Health Research Authority (HRA).

In order to provide reliable risk estimates of relatively rare conditions in a specific group of children born after ART, large, well-designed cohort studies, preferably including families, are necessary. However, as it may not always be appropriate to contact these families directly due to the very personal nature of the treatments involved, access to national-level electronically linked data on cohorts such as the current one provides a cost-effective, non-intrusive and comprehensive way to answer sophisticated research questions in a wide range of areas.

### Strengths and limitations

The main strength of this study lies in the meticulous linkage of robust, routinely collected administrative health data to yield a large cohort that is nationally unique and complements other similar linked cohorts, such as the Nordic linked cohorts,[19] thus increasing the generalisability, accuracy and precision of results from subsequent analyses. The research value and quality and consistency of the HFEA cohort 1992–2009 data have been considerably enhanced through linkage to the ONS birth records and HES databases and removal of artefacts and duplicates. Linkage to the hospital admissions and outpatient database provides long-term mortality and morbidity outcome data on offspring and also represents a high-quality cross-sectoral evidence base that can be used for longitudinal research, policy planning and strategic development.

However, there are also several challenges associated with electronic data linkage studies in general, and Harron et al[72] summarised these into three groups, namely those pertaining to the (1) the data linkage environment and privacy preservation; (2) the linkage process itself, which includes data preparation linkage methods; and (3) the linkage quality and potential bias in linked data. In the context of the current study, limitations in the linkage

process itself included those associated with the method of definition of NC siblings used. The identification of sibling controls would be very sensitive to any errors in linkage, and missed second ART babies would appear as conventional siblings. Parents who had NC children were likely to have been less severely subfertile than parents who did not, and ART children born to the truly infertile would not have conventional siblings. Therefore, the NCS group in this study would have parents with borderline fertility problems or those who developed secondary infertility after the birth of the NC child. For this reason, extensive quality assurance procedures were carried out on the linkage process. With regard to the linkage quality, as this was a bespoke linkage that has never been carried out before, there were no standardised quality assurance measures in place; however, the linkage was extensively checked manually by the team at NHS Digital prior to dissemination for analysis.

The often weak/inaccurate identifier data on the HFEA register and the high threshold for matching used in the current study meant that approximately 23% of children were lost during the linkage process. This is a limitation as almost 100% coverage can be achieved in settings with availability of unique identifying numbers for both mother and child. However, although unavailable for the study period explored here, the HFEA now record both the mother and child's NHS number, suggesting that this loss to follow-up may be avoided in future studies.[73] As HES monitoring data is only available from 1997, children born to women who underwent ART prior to this could not be linked to any hospital records, thus limiting our ability to examine health outcomes in those born before 1997 and exploring the effects of changes in ART techniques on health over that period. However, comparison of maternal age between the unlinked and linked records showed no substantive differences (online supplemental table S8). Unfortunately, unavailability of data prevented us from carrying out further examination and comparison of the educational levels and socioeconomic profile of unlinked records. Nevertheless, linkage of cases born before 1997 to their birth records is still valuable as it may allow investigation of outcomes available on the birth register in the future. Moreover, although these data have been used extensively for research purposes, there have been long-standing concerns regarding the quality, completeness and coverage of HES records within health services and the academic community.[74]

The BW analysis provides an example use of these cohorts and the strengths and limitations of sibling comparisons. However, it cannot be considered as an exhaustive analysis as gestational data was not available for NC children and there is limited covariate data in the present dataset. This is mainly because gestation is not recorded on the birth registration dataset and is of poor quality on the maternal arm of a child's HES record (>50% missing). As a result, although the quality of such data for the ART group (accessed from the HFEA register) was good, it was not available for the NCS and NCP control groups. Further, ART treatment data is available on the HFEA register, which was not included in the current datasets, and these could be extracted for future studies. In particular, the developmental stage of the embryo at transfer and number of embryos transferred are important determinants of treatment success rate (ie, the probability of a child appearing on the HFEA birth register) and also gestation and BW.[37 39 65] Unfortunately, there was no way to systematically identify and account for children born after ART who emigrated or children born to mothers who lived in the England and Wales but travelled abroad for ART treatment, although European estimates of cross-border reproductive care activity suggest that the number of children to whom either of these applies is be small.[75] Due to lack of maternal data, it was not feasible to identify and exclude NCP controls that were part of a triplet or higher order birth. However, the number of such individuals is estimated to be small as the incidence of naturally occurring triplets is in the range of 1:5000–1:9000.[44]

Lastly, an important limitation of the current cohort is that it finishes in 2009. Given that ART technologies have advanced rapidly in the last 10 years, there is an urgent need to continue to prospectively monitor the next cohort of children from 2010 onwards. Two recent UK studies[56 65] have suggested that BWs from both fresh and frozen transfers are increasing with time and may be different in the post 2010 cohort, suggesting that there may potentially be differences (of unknown origin) in the health outcomes of babies born more recently.

## COLLABORATION

The current study carried out a bespoke record linkage that generated a new child cohort for use in exploring the relationship between conception via ART and short-term and long-term health outcomes in offspring and enabling future linkage studies to other developmental and health data sources.

On completion of our further planned analyses, access to the linked dataset will be controlled by HFEA and NHS Digital and specific ethical approval will be required along with approval from the CAG of the HRA.

**Author affiliations**
[1]Population, Policy & Practice Department, UCL Great Ormond Street Institute of Child Health Population Policy and Practice, London, UK
[2]Division of Population Health, Health Services Research & Primary Care, University of Manchester, Manchester, UK
[3]Department of Education, University of Oxford, Oxford, UK
[4]Department of Reproductive Medicine, St Mary's Hospital, Manchester, UK
[5]Department of Obstetrics & Gynaecology, School of Medicine, Dentistry & Nursing, Reproductive & Maternal Medicine, University of Glasgow, Glasgow, UK
[6]MRC Integrative Epidemiology Unit, University of Bristol Bristol Population Health Science Institute, Bristol, UK
[7]Department of Obstetrics, Gynecology, and Reproductive Biology, College of Human Medicine, Michigan State University, East Lansing, Michigan, USA

**Contributors** In line with the ICMJE authorship guidelines, AS, SAR, DRB, SMN, BL and DL made substantial contributions to the conception or design of the work. AGS

and MP made substantial contributions to the acquisition of data. MP undertook data preparation and provided statistical analysis, with SAR and JG providing statistical oversight. MP, SAR, JG, DRB, SMN, DL, BL and AS made substantial contributions to the interpretation of data. All authors contributed to the drafting of the manuscript and/or the revising of the manuscript. All authors have given final approval of the version to be published and agree to its accuracy.

**Funding** This work was supported by the Medical Research Council (Grant number MR/L020335/1).

**Competing interests** None declared.

**Patient consent for publication** Not required.

**Ethics approval** Ethical approval and Section 251 support were obtained from the NHS Research Ethics Committee and Confidentiality Advisory Group, respectively. Additional data access permissions were sought from the HFEA Register Research Panel, ONS Micro-Data Release board and the NHS Digital Medical Register. All researchers with data access underwent NHS Digital Data Security Awareness and ONS Safe Researcher accreditation.

**Provenance and peer review** Not commissioned; externally peer reviewed.

**Data availability statement** Data may be obtained from a third party and are not publicly available. Deidentified linked cohort data can be accessed from the Human Fertilisation and Embryology Authority and NHS Digital where it will be held with restricted access. Specific ethical approval from the Research Ethics Committee (REC) and the Confidentiality Advisory Group (CAG) of the Health Research Authority (HRA) will be required for access.

**ORCID iDs**
Mitana Purkayastha http://orcid.org/0000-0002-1870-8838
Julian Gardiner http://orcid.org/0000-0001-8988-3812

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
