## [Reviewer comments · BMJ Open]

ARTICLE DETAILS

TITLE (PROVISIONAL)	Cohort Profile: A national, population based cohort of children born after assisted conception in the UK (1992-2009) - Methodology and birthweight analysis
AUTHORS	Purkayastha, Mitana; Roberts, Stephen; Gardiner, Julian; Brison, Daniel; Nelson, Scott; Lawlor, Deborah; Luke, Barbara; Sutcliffe, Alastair

VERSION 1 – REVIEW

REVIEWER	Pinborg, Anja
REVIEW RETURNED	30-Mar-2021

GENERAL COMMENTS	General comments This is a well written cohort profile paper with the primary aim of describing the generation of a large cohort of children born after assisted conception based on the national HFEA (Human Fertilisation Embryology Authority) register in the UK. Several papers have already been published on perinatal outcomes originating from HFEA data, but primarily on perinatal outcomes. With this new cohort adding data from UK Hospital Episode Statistics also long-term follow-up of the children is made possible. This will create one of the largest cohorts of ART children worldwide in line with the Nordic CoNARTas database with the possibility of long-term follow-up. Something which is worldwide unique. With 8 million children born worldwide after ART this cohort can gain huge impact by exploring the long term health of ART children. Thus, the paper is publication worthy. However, there are some concerns that need to be addressed before the paper can be approved for publication. • How many children are estimated to be born after ART in UK based on the number of performed ART cycles and are all these included in the HFEA register (page 5: 0,5% in 1992 and 2% in 2017)? Or is there any lack of deliveries? The authors identify 75,348 ART deliveries but is that the full number or how big a proportion is not registered in HFEA? How and whom register deliveries in HFEA is that reported from the clinics and is there any validation of the quantity and quality of this reporting.• In all 63,877 children have follow-up data, why not the rest? Is there any selection bias in the lost to follow-up? Same maternal age, educational level etc?• The UK Hospital Episode Statistics database covers NHS funded hospitals. How big a proportion of UK hospitals are funded by
--

	NHS? The authors should estimate how many hospital visits or admissions that will not be registered by HES. May this lack of registration cause any bias to the study? When children are admitted is it then typically in NHS hospitals.  • How are diagnosed codes registered in HES as ICD-10, ICD-8 codes or which diagnosis classification system is used? • Has there been any validation of the rather complicated identification system of ART children based on mURN, DOB, BW of child, sex of child, forename of mother, surname of mother etc. It seems that about 25% of the children are lost in the identification process, the authors should discuss this? The authors should discuss that in settings with unique PIN codes on mothers and children, full linkage with almost 100% coverage is possible and this loss to follow-up does not occur. Which is a limitation in this UK cohort. Is there any selection bias here i.e. children with very low BW, prematurely born children, stillborn children etc. may be less likely to be registered in the HFEA database? • Many studies have already looked at BW in children from Frozen embryo transfer (FET) vs. fresh embryo transfer and natural conception also in sibling designs. As the authors emphasize data in this study are not new but results are valid based on the existing literature as they point in the same direction as previous studies. What would make the results a lot more interesting would be if the authors may add if the frozen transfer is performed after artificial cycle or natural cycle FET? Or if the authors could add data on preeclampsia in the cohorts. It may be that the question of LGA in FET babies is more linked to the endometrial preparation procedure than the cryotechniques per se. This also fits well into the finding that the BW difference in FET vs. the two other groups in the sibling design is more pronounced than what is seen for the fresh embryo transfer vs. NC. Specific comments Page 7 line 18: What is IMD? Please spell out.
--	---

REVIEWER	Kirby, Russell
REVIEW RETURNED	07-Apr-2021

GENERAL COMMENTS	This paper describes the methods for the creation of a longitudinal cohort of children conceived through ART with birth from 1992-2009 in the UK. The manuscript is very detailed and well written, and my comments are largely editorial in nature. The supplementary materials give considerable useful information for researchers considering the development of similar cohorts in other nationalities. The largest concern has to do with the examples. Since these constitute empirical investigations, the authors should include more details concerning hypotheses, inclusion and exclusion criteria for each study, and following the results some consideration of insights gained about the larger cohort from conducting these studies.
---

	In the section on strengths and limitations which appears on the page after the abstract, what are the 'two approaches' referred to in 4th bullet point? There is no mention in the abstract, so a reader will not know what this refers to. In discussing the results of the linkage, tell the reader more about linkage results (ie p 12). Do the 23% of unlinked records differ in any manner from those that were linked? There are some standard methods for evaluating linkage processes, consider providing more detail here. Throughout, the tables need additional editing. First, by convention (at least in this reviewer's experience) percents are expressed to a single decimal. If two are desired, present the data per 1000 cases. Second, the term 'multiplicity' is not typically used in perinatal epidemiologic research (at least in North America). We use the word 'plurality' to indicate how many fetuses there were in a pregnancy. Third, some column headings are incomplete, for example Table 2 which has columns headed with an open but not a closed parens. Fourth, although the authors may have intended to include notes at the bottom of each table, these are not included in the manuscript. All abbreviations (ie BW, ART, anything abbreviated) should be spelled out in notes. Also there are several items with asterisks in the body of the tables, but no notes defining these. Finally, all tables need more full, descriptive titles, and for the analytical tables, an indication of whether results are crude or adjusted, and if the latter, what covariates were adjusted. All in all this is a very worthwhile contribution, but requires some additional clarification and editing.
--	---

VERSION 1 – AUTHOR RESPONSE

Reviewer: 1

Dr. Anja Pinborg, Rigshospitalet, University of Copenhagen

Comments to the Author:

Comment: This is a well written cohort profile paper with the primary aim of describing the generation of a large cohort of children born after assisted conception based on the national HFEA (Human Fertilisation Embryology Authority) register in the UK. Several papers have already been published on perinatal outcomes originating from HFEA data, but primarily on perinatal outcomes. With this new cohort adding data from UK Hospital Episode Statistics also long-term follow-up of the children is made possible. This will create one of the largest cohorts of ART children worldwide in line with the Nordic CoNARTas database with the possibility of long-term follow-up. Something which is worldwide unique. With 8 million children born worldwide after ART this cohort can gain huge impact by exploring the long term health of ART children.

Thus, the paper is publication worthy. However, there are some concerns that need to be addressed before the paper can be approved for publication.

How many children are estimated to be born after ART in UK based on the number of performed ART cycles and are all these included in the HFEA register (page 5: 0.5% in 1992 and 2% in 2017)? Or is there any lack of deliveries? The authors identify 75,348 ART deliveries but is that the full number or how big a proportion is not registered in HFEA? How and whom register deliveries in HFEA is that reported from the clinics and is there any validation of the quantity and quality of this reporting.

Response: Thank you for your comment. All licensed fertility clinics in the UK are required by law to provide information to the HFEA about treatments they carry out and their outcomes, ensuring high levels of data completeness. As of 07/01/2020, only 1500 birth outcomes for cycles carried out in 2018 had not been reported, almost all of which were likely to be from overseas patients who had ART in the UK and returned home for the delivery thus making it difficult for clinics to follow-up on outcomes. Regular quality assurance measures include manual validation of data submissions from each clinic to the HFEA; regular quality assurance checks on data through the process by which clinics are inspected and licensed by the HFEA; publication of non-compliances with data quality issues in inspection reports; and, where relevant, review of quality reports and auditing of clinics with irregular data submissions.

The following text has been added to the manuscript:

Introduction, page 5: "In the United Kingdom, birth rates from IVF treatment have increased by over 85% since 1991, with around one in three treatment cycles now resulting in a birth for patients under 35 (28)." Databases used section, page 6: "All licensed fertility clinics in the UK are required by law to provide information to the HFEA about treatments they carry out and their outcomes, ensuring high levels of data completeness. Only 1500 ART outcomes had not been reported as of 07/01/2020, almost all of which were likely to be from overseas patients who had ART in the UK and returned home for the delivery making it difficult for clinics to follow up on outcomes (42). Regular quality assurance checks including manual validation of data submissions; regular quality assurance checks on data through the inspection process; publication of non-compliances with data quality issues in inspection reports; and, where relevant, review of quality reports and auditing of clinics with irregular data submissions are also carried out (43)."

Comment: In all 63,877 children have follow-up data, why not the rest? Is there any selection bias in the lost to follow-up? Same maternal age, educational level etc?

Response: Thank you for your valuable comment. Although the HES data warehouse has been operating since 1990, linkage of the HES admitted patient care (APC) episodes longitudinally to the same individuals only commenced in 1997/98 when the patient's NHS number became a mandated return from hospitals. Furthermore, linkage of this APC dataset to other HES datasets such as outpatient, A & E, and adult critical care only commenced much later in 2003/2004, 2007/2008 and 2008/2009, respectively. As a result, only 63,877 children out of the 75348 identified could be linked to follow up health data.

Comparison of maternal age between the unlinked and linked records showed no substantive differences. Unfortunately, unavailability of data prevented us from examining and comparing the educational levels and socioeconomic profile of the unlinked records.

The following text has been added to the 'Discussion' section on page 22 and a table showing the distribution of maternal age in the unlinked records (given below) has been added as a supplementary file.

"However, comparison of maternal age between the unlinked and linked records showed no substantive differences (Supplementary table S8). Unfortunately, unavailability of data prevented us from carrying out further examination and comparison of the educational levels and socioeconomic profile of unlinked records."

Comment: The UK Hospital Episode Statistics database covers NHS funded hospitals. How big a

proportion of UK hospitals are funded by NHS? The authors should estimate how many hospital visits or admissions that will not be registered by HES. May this lack of registration cause any bias to the study? When children are admitted is it then typically in NHS hospitals.

Response: Thank you for your comment. HES data covers all NHS Clinical Commissioning Groups (CCGs) in England, including private patients treated in NHS hospitals, patients resident outside of England, and care delivered by treatment centers [including those in the independent (private) sector] funded by the NHS. It is estimated that 98–99% of hospital activity in England is funded by the NHS. The HES admitted patient care database covers all births in NHS hospitals, representing approximately 97.3% of births in England, thus making creation of nationally representative birth cohorts possible. Based on these facts we believe that the cohort will capture the vast majority of outcomes in couples who became pregnant.

The following text has been included in the 'Databases used- Hospital Episode Statistics' section on page 7.

"HES data covers all NHS Clinical Commissioning Groups (CCGs) in England, including private patients treated in NHS hospitals, patients resident outside of England, and care delivered by treatment centers (including those in the independent sector) funded by the NHS (45). Approximately 98–99% of hospital activity in England are estimated to be funded by the NHS (46). Moreover, the HES admitted patient care database covers all births in NHS hospitals, representing approximately 97.3% of births in England, thus enabling creation of nationally representative birth cohorts (47, 48)."

Comment: How are diagnosed codes registered in HES as ICD-10, ICD-8 codes or which diagnosis classification system is used?

Response: Thank you for your comment. HES diagnoses are coded using the International Classification of Diseases version 10 (ICD-10). ICD-9 was used between April 1989 and March 1995.

The following text has been included in the 'Databases used- Hospital Episode Statistics' section on page 7.

"HES diagnoses were coded using the International Classification of Diseases version 9 (ICD-9) between April 1989 and March 1995 and ICD-10 thereafter."

Comment: Has there been any validation of the rather complicated identification system of ART children based on mURN, DOB, BW of child, sex of child, forename of mother, surname of mother etc. It seems that about 25% of the children are lost in the identification process, the authors should discuss this? The authors should discuss that in settings with unique PIN codes on mothers and children, full linkage with almost 100% coverage is possible and this loss to follow-up does not occur. Which is a limitation in this UK cohort. Is there any selection bias here i.e. children with very low BW, prematurely born children, stillborn children etc. may be less likely to be registered in the HFEA database?

Response: Prior to commencement of this study, the validity of the linkage was tested using a feasibility pilot study involving a sub-group of the main cohort consisting of births in the year from 1st January to 31st December 1998. The results of this pilot study showed that the false positivity rates were very low (<0.0002%, unpublished data).

Matching was done using SQL® using exact matching (design to be inclusive of all potential matches) followed by probabilistic matching using Jaro Winkler® software. A clash was defined as a complete incompatibility of information between databases for a particular variable, with no clashes deemed as being a match. The HFEA were asked for further information in case of one clash, and more than one clash was considered to be a failed match. In the current study, the main linkage was carried out using a very high threshold for matching, and the weak/inaccurate identifiers in the HFEA register often resulted

in matches that failed to pass this threshold. The majority of unlinked records could not be found due to clashes in parental identifiers and child date of birth.

As the main linkage carried out in this study was bespoke in nature, particularly with regard to identification of the naturally conceived siblings, and had not been carried out before, standardized quality assurance measures could not be applied and instead extensive manual checking of the linkage was implemented. Although we do not have data on the demographic profile of the unlinked records ourselves, we have received verbal confirmation from staff at NHS Digital that it was unlikely that the unmatched records would differ significantly from the linked cohort. Moreover, due to the mandatory nature of reporting all ART cycles carried out in the UK to the HFEA, it is unlikely that the unlinked records differed significantly from those that were linked, thus minimizing the risk of selection bias.

The following text has now been added to the paper:

Results section on page 12: "A feasibility pilot study testing the validity of the linkage carried out in this study showed that the false positivity rates were very low (<0.0002%, unpublished data). Matching was done using SQL® using exact matching (design to be inclusive of all potential matches) followed by probabilistic matching using Jaro Winkler® software. A clash was defined as a complete incompatibility of information between databases for a particular variable, with no clashes deemed as being a match. The HFEA were asked for further information in case of one clash, and more than one clash was considered to be a failed match. In the current study, the main linkage was carried out using a very high threshold for matching, and the weak/inaccurate identifiers in the HFEA register often resulted in matches that failed to pass this threshold. The majority of unlinked records could not be found due to clashes in parental identifiers and child date of birth.

Moreover, as the main linkage carried out in this study was bespoke in nature, particularly with regard to identification of the naturally conceived siblings, and had not been carried out before, standardized quality assurance measures could not be applied and instead extensive manual checking of the linkage was implemented. However, due to the mandatory nature of reporting all ART cycles carried out in the UK to the HFEA, it is unlikely that the unlinked records differed significantly from those that were linked, thus minimizing the risk of selection bias."

Discussion section on page 22: "The often weak/inaccurate identifier data on the HFEA register and the high threshold for matching used in the current study meant that approximately 23% of children were lost during the linkage process. This is a limitation as almost 100% coverage can be achieved in settings with availability of unique identifying numbers for both mother and child. However, although unavailable for the study period explored here, the HFEA now record both the mother and child's NHS number, suggesting that this loss to follow-up may be avoided in future studies (73)."

Comment: Many studies have already looked at BW in children from frozen embryo transfer (FET) vs. fresh embryo transfer and natural conception also in sibling designs. As the authors emphasize data in this study are not new but results are valid based on the existing literature as they point in the same direction as previous studies. What would make the results a lot more interesting would be if the authors may add if the frozen transfer is performed after artificial cycle or natural cycle FET? Or if the authors could add data on preeclampsia in the cohorts. It may be that the question of LGA in FET babies is more linked to the endometrial preparation procedure than the cryotechniques per se. This also fits well into the finding that the BW difference in FET vs. the two other groups in the sibling design is more pronounced than what is seen for the fresh embryo transfer vs. NC.

Response: Thank you for your valuable suggestion. While this further analysis would indeed be very interesting, unfortunately lack of data in the HFEA register on the type of FET and preeclampsia

outcomes prevents us from carrying this out. However, a recent study by Castillo et al using more detailed clinic data found no association between the type of FET and birthweight.

The following text has been added to the 'Discussion' section on page 20:

“Although lack of appropriate data on the HFEA register prevented exploration of these effects in the current analysis, a recent UK cohort study by Castillo et al using more detailed clinic data reported no association between type of FET and birthweight (65).”

Specific comment: Page 7 line 18: What is IMD? Please spell out.

Response: Thank you for your comment. The abbreviation is now defined and the text on page 7 now reads:

“Although the current paper only uses demographic data [ethnicity; UK census-derived Index of Multiple Deprivation (IMD), the official measure of relative deprivation for small areas or neighborhoods in the UK (51)] and BW from HES to carry out an exemplar analysis, further analyses of longitudinal health outcomes in these cohorts are already underway and will be published shortly.”

Reviewer: 2

Dr. Russell Kirby, University of South Florida

Comments to the Author:

This paper describes the methods for the creation of a longitudinal cohort of children conceived through ART with birth from 1992-2009 in the UK. The manuscript is very detailed and well written, and my comments are largely editorial in nature. The supplementary materials give considerable useful information for researchers considering the development of similar cohorts in other nationalities.

Comment: The largest concern has to do with the examples. Since these constitute empirical investigations, the authors should include more details concerning hypotheses, inclusion and exclusion criteria for each study, and following the results some consideration of insights gained about the larger cohort from conducting these studies.

Response:

Thank you for your valuable comment. The following text has been added to the paper:

Page 6: “The hypothesis being tested was that children born after frozen embryo transfers would be heavier and those born after fresh embryo transfers would be lighter than those that were naturally conceived.”

The inclusion/exclusion criteria have been discussed under the corresponding section on page 8. To make this explicit we have amended the following text in the 'Exemplar analysis' section on page 11 to read:

“An exemplar analysis examining the effect of fresh and frozen embryo transfer (each compared to NC) on singleton BW was carried out using the HFEA-ONS cohort and HFEA-ONS-HES sub-cohort (inclusion/exclusion criteria discussed previously).”

The wider implications of the findings from the exemplar analysis have been discussed in the Discussion section on pages 20 and 21.

Comment: In the section on strengths and limitations which appears on the page after the abstract, what are the 'two approaches' referred to in 4th bullet point? There is no mention in the abstract, so a reader will not know what this refers to.

Response: Thank you for your valuable comment.

The 4th bullet point of the Strengths and Limitations section on page 3 has been amended to read.

“Comparison of findings between the two approaches (ART vs naturally conceived population controls & ART vs naturally conceived siblings) mentioned above increases confidence in findings”.

Comment: In discussing the results of the linkage, tell the reader more about linkage results (ie p 12). Do the 23% of unlinked records differ in any manner from those that were linked? There are some standard methods for evaluating linkage processes, consider providing more detail here.

Response: Prior to commencement of this study, the validity of the linkage was tested using a feasibility pilot study involving a sub-group of the main cohort consisting of births in the year from 1st January to 31st December 1998. The results of this pilot study showed that the false positivity rates were very low (<0.0002%, unpublished data).

The main linkage was carried out using a very high threshold for matching, and the weak/inaccurate identifiers in the HFEA register often resulted in matches that failed to pass this threshold. The majority of the unlinked records in this study could not be found due to clashes in parental identifiers. As the main linkage carried out in this study was bespoke in nature, particularly with regard to identification of the naturally conceived siblings, and had not been carried out before, standardized quality assurance measures could not be applied and instead extensive manual checking of the linkage was implemented. Although we do not have data on the demographic profile of the unlinked records ourselves, we have received verbal confirmation from staff at NHS Digital that it was unlikely that the unmatched records would differ significantly from the linked cohort. Moreover, due to the mandatory nature of reporting all ART cycles carried out in the UK to the HFEA, it is unlikely that the unlinked records differed significantly from those that were linked, thus minimizing the risk of selection bias.

The following text has now been added to the paper:

Results section on page 12: “A feasibility pilot study testing the validity of the linkage carried out in this study showed that the false positivity rates were very low (<0.0002%, unpublished data). Matching was done using SQL® using exact matching (design to be inclusive of all potential matches) followed by probabilistic matching using Jaro Winkler® software. A clash was defined as a complete incompatibility of information between databases for a particular variable, with no clashes deemed as being a match. The HFEA were asked for further information in case of one clash, and more than one clash was considered to be a failed match. In the current study, the main linkage was carried out using a very high threshold for matching, and the weak/inaccurate identifiers in the HFEA register often resulted in matches that failed to pass this threshold. The majority of unlinked records could not be found due to clashes in parental identifiers and child date of birth.

Moreover, as the main linkage carried out in this study was bespoke in nature, particularly with regard to identification of the naturally conceived siblings, and had not been carried out before, standardized quality assurance measures could not be applied and instead extensive manual checking of the linkage was implemented. However, due to the mandatory nature of reporting all ART cycles carried out in the UK to the HFEA, it is unlikely that the unlinked records differed significantly from those that were linked, thus minimizing the risk of selection bias.”

Discussion section on page 22: “The often weak/inaccurate identifier data on the HFEA register and the high threshold for matching used in the current study meant that approximately 23% of children were lost during the linkage process. This is a limitation as almost 100% coverage can be achieved in settings with availability of unique identifying numbers for both mother and child. However, although unavailable for the study period explored here, the HFEA now record both the mother and child’s NHS number, suggesting that this loss to follow-up may be avoided in future studies.”

Comment: Throughout, the tables need additional editing. First, by convention (at least in this reviewer's experience) percents are expressed to a single decimal. If two are desired, present the data per 1000 cases.

Second, the term 'multiplicity' is not typically used in perinatal epidemiologic research (at least in North America). We use the word 'plurality' to indicate how many fetuses there were in a pregnancy.

Third, some column headings are incomplete, for example Table 2 which has columns headed with an open but not a closed parens.

Fourth, although the authors may have intended to include notes at the bottom of each table, these are not included in the manuscript. All abbreviations (ie BW, ART, anything abbreviated) should be spelled out in notes. Also there are several items with asterisks in the body of the tables, but no notes defining these.

Finally, all tables need more full, descriptive titles, and for the analytical tables, an indication of whether results are crude or adjusted, and if the latter, what covariates were adjusted.

All in all this is a very worthwhile contribution, but requires some additional clarification and editing

Response: Thank you for your valuable comment. The recommended changes have now been incorporated throughout the paper.